# On the Search for the Galactic PeVatrons by Means of Gamma-Ray Astronomy

**Sabrina Casanova** 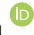

Instytut Fizyki Jądrowej, Polskiej Akademii Nauk, 31-342 Krakow, Poland; sabrina.casanova@ifj.edu.pl

**Abstract:** Cosmic rays are ultra-relativistic particles that slam into the atmosphere from all directions in the sky. Gamma rays emitted when cosmic rays interact with Galactic gas and radiation fields are a powerful tool to investigate their origin. Many candidate CR sources have been discovered in GeV-to-PeV gamma rays. However, the major contributors to the CR population, especially at the highest energies, are still unknown. We give here a state of the art report on the search for the sources of Galactic cosmic rays by means of gamma-ray astronomical methods.

**Keywords:** gamma-ray astronomy; high energy astrophysics; interstellar medium





## 1. Introduction

Cosmic rays (CRs) are the highly energetic particles that impinge on the Earth from all directions in the sky and carry above 1 GeV and an energy density of about 1 eV/cm$^3$, which is comparable to the energy density of the Galactic magnetic fields, radiation fields, and turbulent motions of the interstellar gas. Since their discovery in 1912 by Viktor Hess, the search for the astrophysical sources capable of accelerating these particles has been one of the main research interests in high energy astrophysics [1].

The CR population measured at Earth is mostly composed of protons and heavier nuclei, electrons, and positrons representing roughly 1% of the flux at GeV energies. The CR spectrum, which extends over a vast range of energies from $10^9$ to $10^{21}$ eV, has a prominent feature, the so-called knee, at a few PeV (PeV = $10^{15}$ eV). Thanks to the increasing precision of CR measurements, a hardening of the proton and heavy nuclei spectra at about 200 GeV/n in rigidity has been discovered, with the different CR species showing discrepant spectral hardening indices. The knee is thought to mark the change from Galactic into extra-Galactic CRs, but could be also be a transport feature or could be due to the convolution of different source spectra. The origin of the hardening in the spectrum, which is confirmed up to 100 TeV [2–4], is also unclear and might be the result of acceleration processes, such as the contribution from discrete sources close to the Solar System or re-acceleration in weak SN shocks, or be caused by transport effects.

Up to several PeVs CRs are believed to have a Galactic origin, accelerated in the so-called *TeVatrons* or *PeVatrons*, the TeV and PeV factories of Galactic CRs, respectively. The nature of the astrophysical sources, which are the major TeVatrons and PeVatrons, is yet to be clarified.

Since CRs are charged particles, they do not point to their acceleration sources because they deflect in the Galactic magnetic fields. It is thus impossible to pin down their sources. A powerful tracer for CR populations distant from the Earth are gamma rays, which are emitted in various interactions between the CRs and their environment at the locations of CR sources and in their vicinity. Gamma-ray signatures of TeVatrons and PeVatrons are expected to unveil the astrophysical sources capable of accelerating CRs up to PeV energies.

For CRs from GeV-to-PeV energies, the dominant emission processes are the decay of neutral pions, $\pi_0$, produced when CR hadrons collide with ambient gas in the ISM, which is commonly called the *hadronic* production mechanism, and inverse Compton (IC) scattering of CR electrons off radiation fields, which is called the *leptonic* production mechanism.

Proton–proton collisions will produce gamma rays of around 10% of the energy of the parent cosmic ray; for instance, several tens to hundreds of TeVs for CRs close to PeVs. Typically, the spectrum of the hadronic radiation at TeV mimics the parental CR spectrum where the proton energy is shifted to a lower energy by a factor of 20–30, so that PeV CRs emit $\gamma$ rays at hundreds of TeVs. The cross-section of the inverse Compton scattering if the electron energy is significantly higher than the target photon energy has been derived by [5–9]. The angle-averaged total cross-section of inverse Compton scattering depends only on the product, $k_0$, of the energies in $m_e c^2$ of the interacting electron, $E$, and of the target photon, $\omega_0$, $k_0 = E\omega_0$. In the non-relativistic regime ($k_0 < 1$), the cross-section approaches the classical Thomson cross-section $\sigma_{IC} \approx \sigma_T (1 - 2 k_0)$. In the ultra-relativistic Klein–Nishina (KN) regime ($k_0 \gg 1$), the cross-section goes as $\sigma_{IC} \approx (3/8) \sigma_T (k_0)^{-1} \ln(4 k_0)$. The energy of the emitted photons, $E_\gamma$, is $\omega_0 \leq E_\gamma \leq \frac{4k_0}{1+4k_0} E$. In the extreme KN regime, the energy of the emitted photon approaches the electron energy.

At MeV–GeV energies, non-thermal electron bremsstrahlung radiation can be an important process if the gas density is high. The same population of electrons IC upscattering low energy photons to GeV and TeV photons emit synchrotron radiation in radio and X-ray bands, respectively. Synchrotron radiation is hereafter considered mainly as an electron cooling mechanism. Synchrotron radiation is, however, emitted also by secondary electrons, products of $\pi^\pm$-decays when protons collide with the ambient gas. These X-rays from secondary electrons can be treated as a *prompt* radiation, which is emitted simultaneously with gamma rays in multi-Tevatron and PeVatron CR sources.

Gamma-ray observations of different astrophysical sources can help to constrain the sites and acceleration mechanisms producing the hadronic part of the cosmic ray flux that we measure at Earth. In the following sections, we will discuss recent progress in the search for the Galactic PeVatrons. Such a search is not complete, as we currently do not have robust observations of a sizable population of the hadronic PeVatrons that could produce the CR knee observed at Earth. While understanding the sources of the local CR flux, the local *fog* (I heard this appropriate term during a presentation by Prof Aharonian), is of paramount interest; understanding the acceleration mechanisms of the fast perfect accelerators capable of injecting particles up to PeV energies is of more general interest. In this context, gamma-ray observations are the best tool to shed light on the physics of the most extreme accelerators in the Galaxy, capable of accelerating particles up to and beyond PeV.

## 2. Young Supernova Remnants

The initial idea that CRs originate from Galactic supernova remnants (SNRs) was formulated in 1964, based on energetic considerations [10]. Supernova explosions, happening at a rate of one event every 30 years in the Galaxy, inject on average $10^{51}$ ergs. Ten percent of the explosion energy is enough to account for the energy in Galactic CRs [10]. According to the *SNR paradigm*, CRs are accelerated in the shells of young SNRs through a mechanism known as diffusive shock acceleration (DSA) in the presence of strong magnetic fields. Diffusive shock acceleration in shell-type supernova remnants provide the adequate conditions for efficient CR acceleration up to at least 100 TeV and predicts that 10% of the kinetic energy available in supernova explosions is transferred to cosmic rays, in accordance with the energetic requirements.

Several SNRs have been studied at GeV and TeV energies, among which are young SNRs like Cas A [11–14], Tycho, [15,16], and RX J1713.7-3946, [17,18]; and middle-aged SNRs like IC433 [19,20], W28 [21,22], and W51C [23]. In these latter cases, the TeV radiation is thought to be produced by runaway CRs interacting with the dense gases of nearby molecular clouds (see below).

Among the young shell-type SNRs observed at TeV energies, RX J1713.7-3946 stands out for its brightness in non-thermal X-ray and TeV gamma rays, and has been deeply studied at various wavelengths. The shell morphology of RX J1713-3946 after 10 years of High Energy Stereoscopic System (H.E.S.S.) data, together with the X-ray contours from

XMM are shown in the top left panel of Figure 1 [17,18,24,25]. The spatially resolved remnant has a shell morphology at TeV energies similar to that seen in X-rays. This similarity tells us that electrons are effectively accelerated in the shell, and that the same population of electrons accelerated up to hundreds of TeVs emits both X-ray synchrotron radiation and TeV radiation through inverse Compton scattering. The remarkable difference between the X-ray and TeV radiation is that the gamma-ray source extends radially beyond the X-ray (see radial profiles in Figure 1). The main shock position and extent of the shell are also visible in the X-ray data. The TeV emission extending further is either due to accelerated particles escaping the acceleration shock region, or particles accelerated in the shock precursor region. As discussed in [18], the same difference between TeVs and the X-ray shell can be seen when dividing the data set into energy bins, namely for all energies, for E < 1 TeV, 1 < E < 3 TeV, and E > 3 TeV.

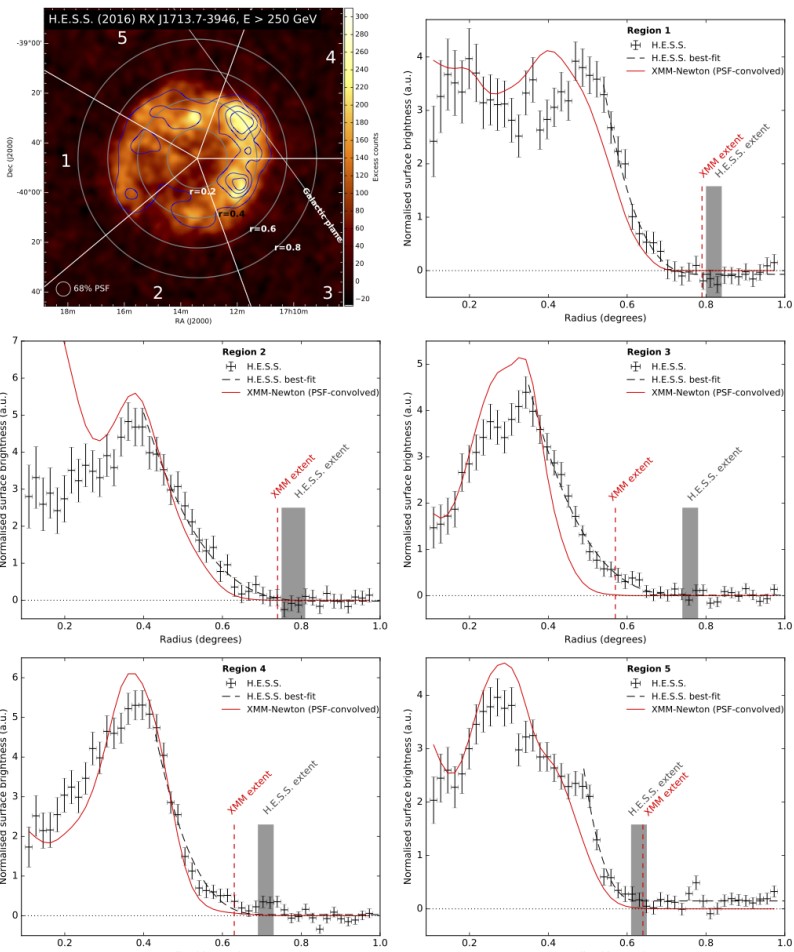

**Figure 1.** The VHE gamma-ray excess map and radial profiles of RX J1713.7-3946 above 250 GeV obtained from 170 h observations by the H.E.S.S.s telescope are shown [18]. Top left: the High Energy Stereoscopic System (H.E.S.S.) gamma-ray count map (E > 250 GeV) is shown with XMMs X-ray contours (1-10 keV, smoothed with the H.E.S.S. PSF) overlaid. The five regions used to compare the gamma-ray and X-ray data are indicated along with concentric circles (dashed grey lines) with radii of 0.2° to 0.8° and centered at the SN location. The radial profiles of region 1 to region 5 are shown in the top right panel, and in the middle and bottom panels. Black crosses correspond to the TeV profiles extracted from the H.E.S.S. map. The red lines are X-ray profiles extracted from the XMM map convolved with the H.E.S.S. PSF. The relative normalization between the H.E.S.S.s and XMMs profiles is chosen such that for regions 1, 2, and 4, the integral in [0.3°, 0.7°] is the same for regions 3 and 5 in [0.2°, 0.7°]. The grey shaded area shows the combined statistical and systematic uncertainty band of the radial gamma-ray extension. The vertical dashed red line is the radial X-ray extension.

Protons are likely accelerated in the supernova shell through the same mechanism of accelerating electrons, and the TeV emission could be completely or partially produced by these protons. It is not clear, however, what fraction of energy input of the supernova explosion goes into the acceleration of electrons, and what fraction goes into the acceleration of protons. The broadband shell spectra, shown in Figure 2, can help with estimating the energy input in protons and electrons. As is evident in Figure 2, both hadronic and leptonic emission models can account for the broadband radiation from the shell of RX J1713.7-3946 (*leptonic-hadronic degeneracy*).

The full-remnant spectral energy distribution shown at gamma-ray energies exhibits a hard spectrum in the GeV regime, a flattening between 100 GeV and a few TeV, and an exponential cut-off above 10 TeV. In a hadronic scenario, this spectrum can be fitted with a broken power law proton spectrum with a break at $1.4 \pm 0.5$ TeV and an exponential cut-off at about 100 TeV. For a target density of n = $1/cm^3$, the energy budget above 1 TeV amounting to $(5.80 \pm 0.12) 10^{49}$ erg above 1 TeV is required to explain the measured gamma-ray flux [18].

The observed X-ray and gamma-ray spectrum can be alternatively produced by a broken power law electron population with a break at 2.4 TeV. The magnetic field strength required to reproduce the X-ray and gamma-ray spectra is B = $14.2 \pm 0.2$ μG. If the fit includes the X-ray data, the exponential cut-off of the parent particle spectrum is $65 \pm 7$ TeV, compared to $88.4 \pm 1.2$ TeV. The IC emissions are much more efficient at producing VHE gamma rays than protons via pion decay. A proton spectrum that is about 100 times higher is needed to produce nearly identical gamma-ray curves [18]. Finally, we note that both the energy budget in protons and electrons are dependent upon the magnetic and radiation fields and on the gas density, which are not well constrained.

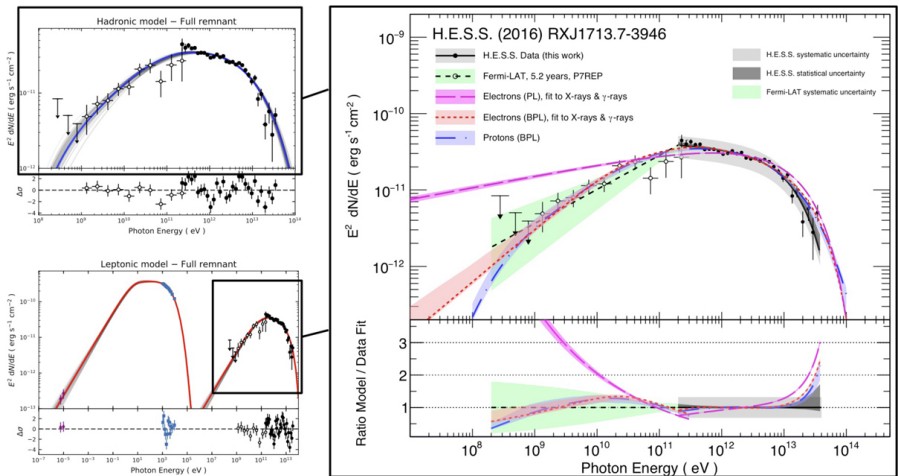

**Figure 2.** The hadronic gamma-ray model and the leptonic gamma-ray model for the SNR RX J1713.7-3946 compared to data, including lower-energy Susaku X-rays and radio data from ASKA [26] are shown in the top and bottom left panels, respectively (Figure from [18]). The shell broadband spectrum, including data from H.E.S.S. and Fermi-LAT gamma-ray data as solid and open circles, is compared to the best fitting models in the right panel. In addition to the best-fit models of an electron broken power law with a cut-off (BPL), a power law without cut-off is also shown for the leptonic model to demonstrate that this model cannot describe the Fermi-LAT gamma-ray data.

Finally, X-ray synchrotron emission with a variability timescale of few years has been observed in the shell of this SNR [24]. The variability of X-rays images on year timescales shows that the electron acceleration in RX J1713-3946 takes place in a strongly magnetized environment with an amplification of the magnetic field by a factor of more than 100 [24]. These amplified magnetic fields, which cool down extremely fast electrons accelerated up to hundreds of TeV, are a necessary condition for the acceleration of protons to energies beyond 100 TeV [24].

SNRs are believed to accelerate the highest energy CRs (up to at least a few PeVs) at the transition between the free expansion and the Sedov phase, within several hundred years after the supernova explosion. During the Sedov phase, the SN shock slows down and the magnetic field intensity decreases, so that the shock cannot confine any longer the most energetic particles, which escape the SNR. Considering the rate of SN explosions in the entire Galaxy (about three per century), the chances to observe a SNR when it is still a *PeVatron* are thus very low, and any proof of emission from PeV CRs even from young SNRs would be challenging to find. Lower energy CRs are confined for longer within the SNRs, but eventually will escape too. CR escape is one of the main reasons for why it is difficult to constrain the maximum acceleration energy and estimate the total power released in CRs by SNRs.

Very young SNRs are clearly primary targets to test the SNR paradigm. The youngest known SNR in the Galaxy is G1.9+0.3, which is only 110 yr old. While its shock speed of 10,000 km/s, estimated through X-ray measurements, is ideal for producing efficient diffusive acceleration, H.E.S.S.s could not detect any evidence of TeV emission, and thus of efficient particle acceleration, in this very young SNR. Although the efficiency of particle acceleration might be low in the very first phases of SNR evolution, this result confirms the difficulty in testing the SNR paradigm.

Molecular clouds (MCs) are regions of the Galaxy, typically a few tens of parsecs in radius, where the density of cold molecular gas is often orders of magnitude higher than elsewhere in the diffuse ISM. Stars are believed to be born in these clouds, and massive stars, which explode as SNRs, live their short lives within their birthplaces, and so within the MCs where they were born [27–29]. Runaway CRs leaving early on the SNRs have plenty of target material to collide with and produce gamma-ray emissions. Giant molecular clouds enhance the gamma-ray emissions produced through the hadronic CR channel. So, the relative contribution of the IC leptonic component of the VHE emission compared to the $\pi^0$-decay gamma rays from specific dense regions is significantly reduced because of the enhanced gas density [28–36]. Enhanced gamma-ray emission from molecular clouds located close to young SNRs can thus provide the first evidence of the parent population of runaway CRs. Depending on the location of massive clouds, on the acceleration history, and on the timescales of the particle escape into the interstellar medium related to the diffusion coefficient, a broad variety of energy distributions of gamma rays is produced—from very hard spectra (much harder than the spectrum of the SNR itself) to very steep ones [30,37,38].

At TeV energies, MAGIC and VERITAS detected extended emissions from the vicinity of the SNR IC 443, with its brightest regions being coincident with the dense cloud material and maser emission. The H.E.S.S. observations have revealed VHE gamma-ray sources in the field of the SNR W28, which positionally coincide well with molecular clouds studied with the Nanten telescope [21]. Such emission correlates quite well with the position of dense and massive molecular clouds, and thus it is often interpreted as the result of hadronic cosmic ray interactions in the dense gas. A re-analysis of Fermi-LAT data outside the middle-aged SNR, W44, revealed clouds of runaway cosmic-rays escaping collectively and anisotropically from the shell along the magnetic field of the remnant [39], as predicted by some theorists [40,41].

To recap, the GeV-to-TeV radiation from young shell-type SNRs, which has been long thought to deliver a solution in the question of the origin of cosmic rays, can be either of hadronic or leptonic origin (*leptonic-hadronic degeneracy*). The energy budget in accelerated electrons and protons, $W_e$ and $W_p$, can be obtained from the gamma-ray luminosity, $L_\gamma$, as $W_e = L_\gamma t_{IC}$ and $W_p = L_\gamma t_{pp}$, respectively. Typically a 1 TeV gamma-ray photon is emitted either by an electron or a proton of about 10 TeV. The cooling time for a 10 TeV electron is $t_{IC} \approx 5 \times 10^4$ yr (for scattering off the CMB photons), while for a proton, the cooling time is $t_{pp} \approx 5 \times 10^7 (n/1 \text{ cm}^{-3})^{-1}$ yr (see e.g., ref. [42]). Thus, the ratio of the luminosity in IC gamma-rays to $\pi^0$-decay gamma-rays is of the order of $10^3 \times (\frac{W_e}{W_p}) \times (\frac{n}{1 \text{ cm}^{-3}})^{-1}$. The leptonic contribution to the emission is thus dominant over

the hadronic one, unless $\frac{W_e}{W_p} << 10^{-3}$, which could be true in SNRs because of the short synchrotron cooling time of electrons [43]. Alternatively, the gas density inside the shell has to be $n >> 1\,\mathrm{cm}^{-3}$, which would, however, have the drawback of slowing down the shock wave, and thus, would prevent efficient acceleration. Finally, in the presence of magnetic field, B, exceeding 10 µG, the electrons accelerated within the SNR are cooled predominantly via synchrotron radiation, and thus, only a small fraction, $w_{MBR}/w_B \approx 0.1(B/10\,\mu\mathrm{G})^{-2}$ (where $w_{MBR}$ and $w_B$ are the energy densities of the microwave background radiation field and of the magnetic field, respectively), is released in IC gamma ray. In fact an amplified magnetic field, exceeding 10 µG, which can be induced by cosmic ray streaming instability in the region upstream of the shock, has been observed in the shell of young SNRs detected at TeV energies.

The power in accelerated protons within SNRs derived from gamma-ray observations depends on the highly uncertain local density n as 1/n, and on the magnetic and radiation fields within the remnant. Thus, it is not possible to prove that the dominant contribution to the observed TeV emissions from these supernova remnants is produced by accelerated protons, or that the population of SNRs in the Galaxy injects the necessary power (about $10^{50}$ erg per supernova event) to sustain the CR population.

## 3. The Search for the Major Galactic CR Factories

The bottom line of all the studies of SNRs conducted at TeV energies is that SNRs have been proven to be efficient accelerators of cosmic ray electrons up to 100 TeV. That protons up to 100 TeV are accelerated in young SNRs seems a reasonable assumption, despite remaining unproven. However, the hypothesis that acceleration of cosmic rays within the shells of SNRs proceeds up to PeV energies has been rejected in all known young SNR emitting TeV photons, as all SNRs detected by Imaging Atmospheric Cherenkov Telescopes (IACTs) show clear cut-offs in the gamma-ray spectra at several TeVs (see Figure 3).

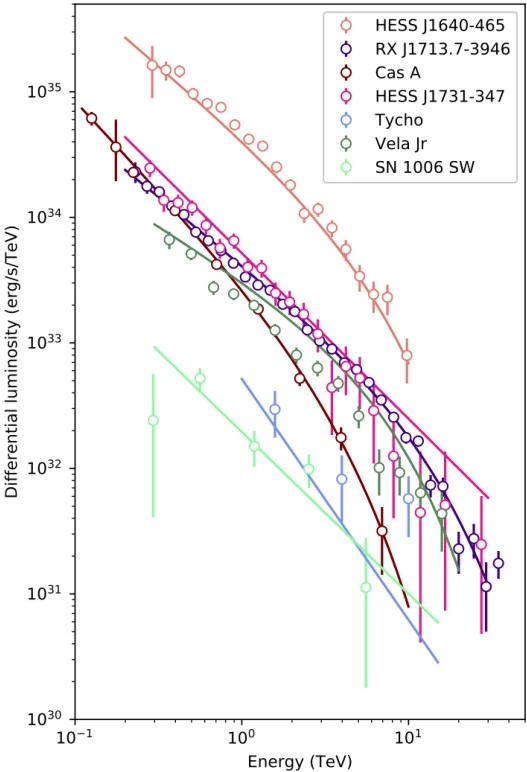

**Figure 3.** Differential luminosity curves for the some of the most commonly studied SNRs at TeV energies. The spectra show cut-offs at several TeV, which point towards efficient acceleration up to few tens–hundreds of TeV. Figure from [44].

Since CRs with energies of up to at least a few PeV ($=10^{15}$ eV), the knee in the CR spectrum, are believed to be of Galactic origin, one or more sources that are able to accelerate particles up to PeV energies, so-called PeVatrons, must inhabit our Galaxy. Some other classes of astrophysical sources, different from SNRs, such as super-bubbles, star-forming regions (see Section 3.2), or remnants of GRBs in our Galaxy, and in particular, the Center of our Galaxy, have long been proposed as effective accelerators of particles and as major contributors to the population of Galactic cosmic rays up to PeV energies [45–53].

### 3.1. The Population of Galactic PeVatrons

The first evidence of a breakthrough in the understanding of the origin of the highest energy CRs in the Galaxy has been the discovery of a powerful PeVatron in the Centre of the Milky Way, which has been associated either with the radio source at the Center of our Galaxy, Sgr A*, or to three powerful star clusters in the Central Molecular Zone (CMZ), the Arches, the Quintuplet, and the Nuclear clusters [44,54,55]. In Figures 4 and 5, the gamma-ray map and the spectrum of the GC diffuse emission are shown, respectively. The gamma-ray distribution follows closely the molecular gas distribution, and can be interpreted as being produced by hadrons under the assumption that the particle diffusion coefficient is space independent. In Figure 5, the spectrum of the central source, HESS J1745-290, has a clear cut-off, while the diffuse emission shows no clear evidence of a cut-off. The interpretation proposed in [54] is that the highest energy cosmic rays accelerated in the central source have already diffused out and reached the dense gas clouds in the Central Molecular Zone. An alternative interpretation of the emission from the Galactic Center sees the diffuse emission from the Galactic Center as being due to a hardening of the CR spectrum in the region, triggered by a different diffusion regime [56,57].

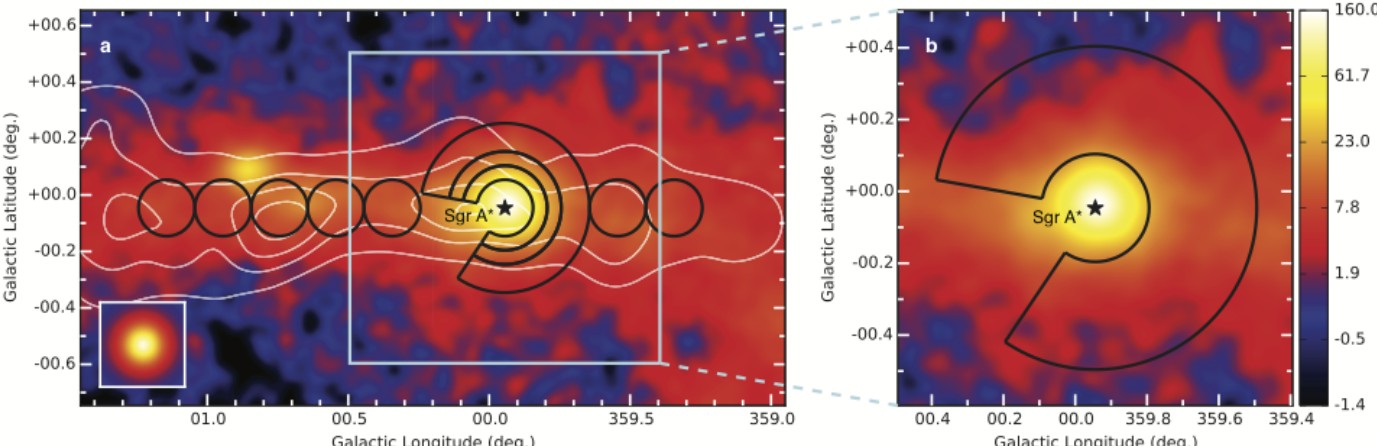

**Figure 4.** (**a**) H.E.S.S. gamma-ray excess map of the Galactic Center region. White contour lines indicate the density distribution of molecular gas traced by its CS line emission in the Central Molecular Zone. (**b**) Zoomed view of the inner 70 pc of the Galactic Centre region, and the contour of the region used to extract the spectrum of the diffuse emission. Figure from [54].

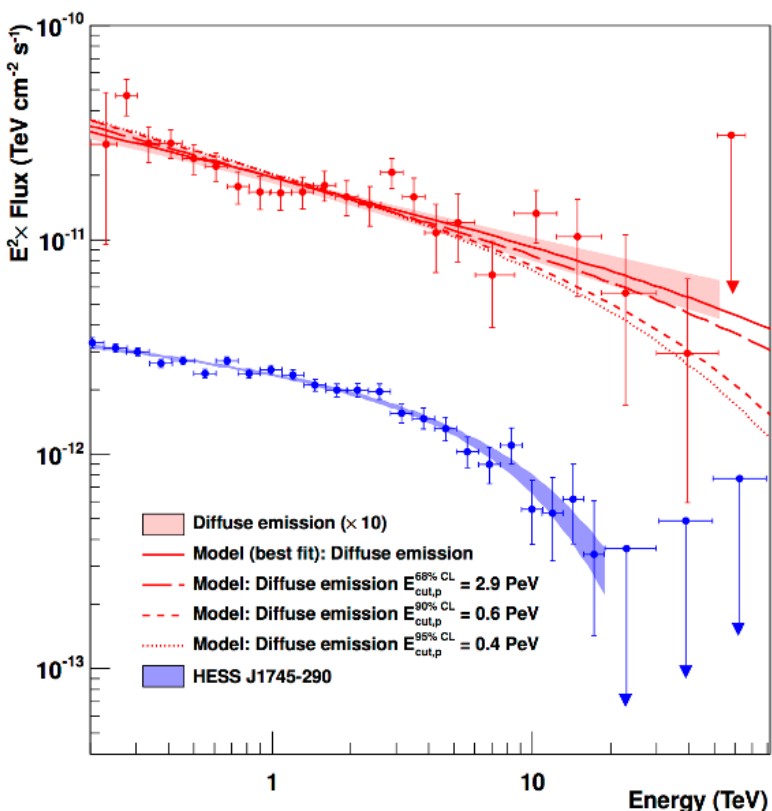

**Figure 5.** VHE gamma-ray spectra of the diffuse emission and HESS J1745-290. The vertical and horizontal error bars show the 1 $\sigma$ statistical error and the bin size, respectively. Arrows represent 2 $\sigma$ flux upper limits. The 1 $\sigma$ confidence bands of the best-fit spectra of the diffuse and HESS J1745-290 are shown in the red and blue shaded areas, respectively. The red lines show the numerical computations, assuming that gamma rays result from the decay of neutral pions produced by proton–proton interactions. The fluxes of the diffuse emission spectra and models are multiplied by 10 to visually separate them from the HESS J1745-290 spectrum. Figure from [54].

Unidentified gamma-ray sources, such as HESS J1641-463 and HESS J1702-420A, extending up to several tens of TeV without evidence of a cut-off in the gamma-ray spectrum, have been discovered by H.E.S.S. in different regions of the Galactic Plane [58–60]. The absence of any break in the spectrum up to several tens of TeV (see Figure 6), and the frequent correlation with molecular clouds, suggests that the dominant mechanism for the emission from these unidentified sources could be hadronic, and in this case, the parental proton spectrum should extend to energies close to the knee. In Figure 6, we show how the parental particle population can be deduced from the TeV emission.

The search for PeVatrons sources has been given a new impulse by the survey of the Galactic Plane at higher photon energies carried out with the HAWC and LHAASO Observatories (see Figures 7 and 8, respectively). HAWC has observed several gamma-ray sources emitting above 56 TeV, with three of them showing emissions continuing to 100 TeV and beyond [61–64]. While one of these sources is likely associated with the SNR, G106.3+2.7, evidence of powerful particle acceleration close to PeV energies has emerged from the Cygnus star forming region [64], and in the vicinity of a 2MASS young star cluster [63].

The LHAASO Collaboration reported recently the detection of more than 530 photons at energies above 100 TeV from 12 ultra-high energy gamma-ray sources, and up to 1.4 PeV from the Cygnus region of the Galaxy [65]. The photon energies reported by the LHAASO Collaboration are so high that the attenuation of these high energy gamma rays, due to the pair production of gamma rays interacting with background photons from both the cosmic

microwave background (CMB) and interstellar radiation fields (ISRF), discussed among others by [66,67], revealed itself thanks to the capability of LHAASO to observe at the highest energies, detect relatively distant sources, and measure the gamma-ray spectrum with relatively small statistical uncertainties.

Another crucial piece of information to clarify the origin of the highest CRs are the HAWC and Tibet Air Shower Array measurements of diffuse emissions up to hundreds of TeV from the Galactic Plane. This radiation, compatible with expectations from an hadronic emission scenario, in which gamma rays originate from the decay of pions produced through the interaction of protons with the Galactic interstellar medium, is a strong evidence that cosmic rays are accelerated beyond PeV energies in our Galaxy and spread over the Galactic disk [68,69].

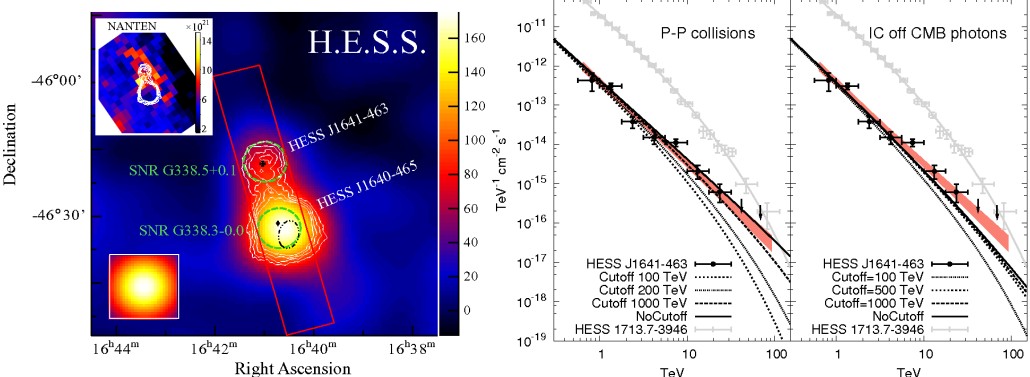

**Figure 6.** *On the left:* The H.E.S.S. emission from the region of the hard spectrum source, HESS J1641-463, hosting the nearby bright SNR, SNR G338.3-0.0, coincident with HESS J1640-465. In the bottom insert, the H.E.S.S. PSF is presented, while the upper insert shows the gas distribution in the region according to the NANTEN CO survey [70]. Cosmic rays leaking from G338.3-0.0 and interacting with dense molecular clouds is the most likely scenario to explain the TeV emission detected from HESS J1641-463. HESS J1641-463 has one of the hardest spectra ever detected in TeV astronomy, extending up to 30 TeV without evidence of a cut-off (see the *right panel*). The differential VHE gamma-ray spectrum of HESS J1641-463 for energies between 0.64 TeV and 100 TeV, together with the expected emission from p–p collisions (left) and IC from CMB photons (right) is shown in the *right panel*. The pink area represents the 1-$\sigma$ confidence region for the fit to a power law model, the red data points to the H.E.S.S. measured photon flux (1-$\sigma$ uncertainties), the arrows the 95% CL upper limits on the flux level, and the black curves the expected emission from the models, assuming different particle energy cut-off values. For comparison, the gray data points and curves represent the archival spectrum and the corresponding best fit model, respectively, of the young SNRRX J1713.7-3946 [71] The two plots are from [58].

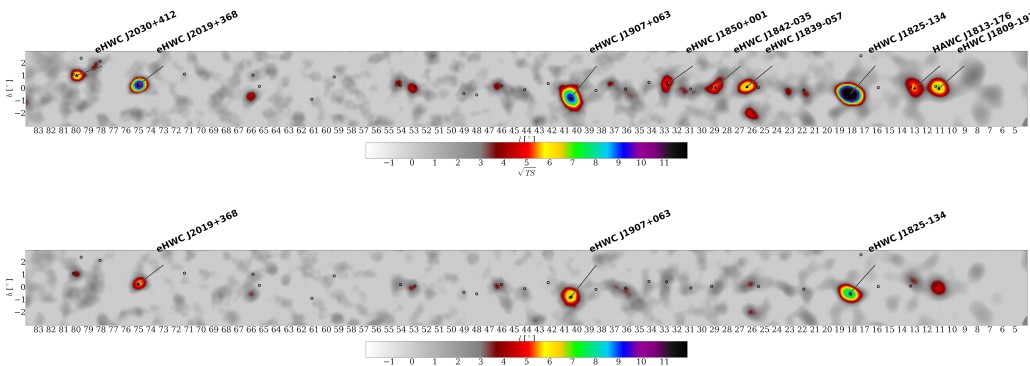

**Figure 7.** TS map of the Galactic plane for photon energies above 56 TeV (top panel) and above 100 TeV (bottom panel). A disk of radius 0.5 degree is assumed as the morphology. Black triangles denote the high energy sources. For comparison, black open circles show sources from the second HAWC catalog [61–64,72].

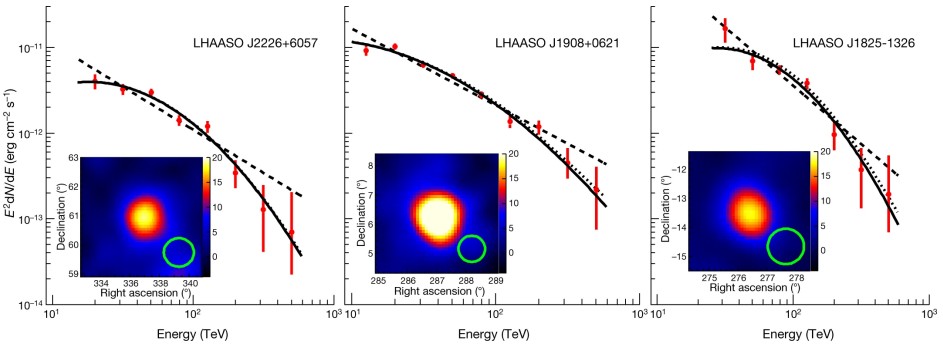

**Figure 8.** Spectral shapes of three of the Galactic Plane sources observed by the LHAASO Collaboration extending up to several hundreds of TeV. Figure from [65].

### 3.2. Star Forming Systems—The Case of the Cygnus Region

Collective stellar winds and SNR shocks in clusters, and associations of massive stars have been long suggested as possible alternatives or additional proton TeVatrons and PeVatrons [44–47,73–75]. Particle acceleration can in fact proceed very efficiently within star clusters and the associations of massive stars, where core-collapse SN progenitor stars and colliding wind binaries spend most of their lives in groups of loosely bound or dense associations. The winds of multiple massive stars in such systems can collide and form collective cluster winds that drive a giant bubble, a so called superbubble, filled with hot ($T = 10^6$ k) and tenuous ($n < 0.01$ cm$^{-3}$) plasma. In these systems, acceleration shocks can be formed at the termination of the stellar cluster wind; also, turbulences in form of MHD fluctuations and weak reflected shocks can build up very efficiently [48–52]. Supernova explosions of massive stars in thin and hot superbubbles can also produce efficient particle acceleration at the boundary of the superbubbles or at MHD turbulence, and further amplify existing MHD turbulence [76,77]. These multiple shocks can result in efficient acceleration even beyond PeV energies [78]. Moreover, turbulences in superbubble interiors can accelerate particles to very high energies, not only through a first-order Fermi mechanism, but also via the second-order Fermi mechanism [50].

Within stellar clusters and massive star associations, the interaction of the accelerated particles with the ambient medium, including often dense molecular clouds or electromagnetic fields leads also to the efficient production of VHE gamma rays. We will here illustrate how the GeV-to-PeV observations of the Cygnus region support the hypothesis that star-forming regions are sites of high energy particle acceleration [64,65,79–86].

The Cygnus region hosts some of the most remarkable star-forming regions in the Milky Way, including Cygnus X at only 1.5 kpc from the Sun, with a total mass in molecular

gas of a few million solar masses, which is at least 10 times the total mass in all other close-by star-forming regions such as Carina or Orion, and a total mechanical stellar wind energy input of $10^{39}$ erg s$^{-1}$. This corresponds to several percent of the kinetic energy input by SNe in the entire Galaxy. Cygnus X hosts many young star clusters and several groups of O- and B-type stars, called OB associations. One of these associations, Cygnus OB2, contains 65 O stars and nearly 500 B stars. These super stars have likely created cavities filled with hot, thin gas surrounded by ridges of cool, dense gas where stars are now forming, which strongly emit at GeV energies, called the Fermi Cocoon superbubble [87].

The TeV counterpart of the Fermi Cocoon has been studied with the ARGO detector up to about 10 TeV, with the HAWC observatory up to at least 200 TeV (see Figure 9), and with LHAASO up to 1.4 PeV [64,65,88]. Cosmic rays up to PeV energies, accelerated in the enclosed star cluster Cyg OB2, likely produce the detected emission. The spectral energy distribution shows a significant softening at a few TeV: this is revealed by the comparison between the ARGO and HAWC data and the Fermi-LAT data. This break in the gamma-ray spectrum hints at a cut-off in the injected CR spectrum, or can possibly be explained as being due to suppressed diffusion in the high-turbulence environment of the Cygnus superbubble, which would confine low-energy particles, whereas higher-energy particles escape. An analysis of the CR density as a function of the distance from the star cluster has revealed at Fermi-LAT energies a distinct signature of the continuous injection of CRs over the cluster lifetime and following diffusion through ISM [44]. The $1/r$ decrement of the CR density with the distance from the star cluster is, in fact, a distinct signature of the continuous injection of CRs over the cluster lifetime and their diffusion through ISM. At TeV energies, the CR energy density as a function of the distance from the center of the emission, roughly coincident with the location of the CygOB2 cluster, is shown is Figure 10. The CR density, as resulting from TeV measurements, is significantly higher than the CR energy density at Earth. It could, however, be produced by both a continuous injection or by a burst-like event happening in the cluster a tenth of a million years ago [64].

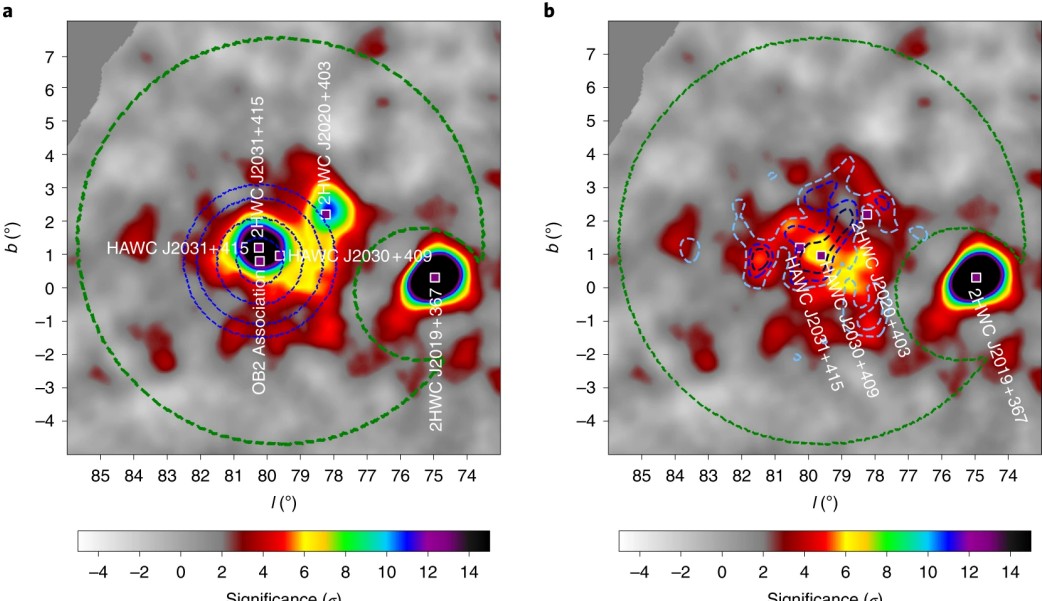

**Figure 9.** (**a**) Significance map of the Cocoon region before, and subtraction of the known sources in Galactic coordinates, latitude, b, and longitude, l. The blue contours are four annuli centered at the OB2 association, where the CR density was estimated. The green contour is the ROI used for the study, which masks the bright source 2HWC J2019+367. (**b**) Significance map of the Cocoon region after subtracting HAWC J2031+415 (PWN) and 2HWC J2020+403 ($\gamma$-Cygni). The light-blue, medium-blue, and dark-blue dashed lines are contours for 0.16, 0.24, and 0.32 photons per 0.1 deg $\times$ 0.1 deg spatial bin, respectively, from Fermi-LAT Cocoon. Figures from [64].

The GeV-to-TeV observations of star forming regions and massive stellar clusters in our Galaxy, which have given new impulse to the gamma-ray research in high energy astrophysics, help to constrain the fraction of mechanical stellar wind energy transferred into relativistic particles and hence gamma rays, studying particle acceleration and propagation in Galactic stellar clusters and superbubbles. Furthermore, high energy phenomena are attracting increasing attention in relation to the life cycle of interstellar matter and star-formation processes. The rate and efficiency of the star formation process depends in fact on the balance between the self-gravity of dense molecular cores and countervailing forces, which act to support the clouds. The most important of these are likely to be thermal pressure, turbulence, and magnetic fields. In order for magnetic support to be effective, a population of ionized particles must be present in the core. Since molecular clouds are opaque to ultraviolet radiation from stars, the main ionizing agent is thought to be low-energy CRs, and the magnetic support of the cloud is critically dependent on this factor.

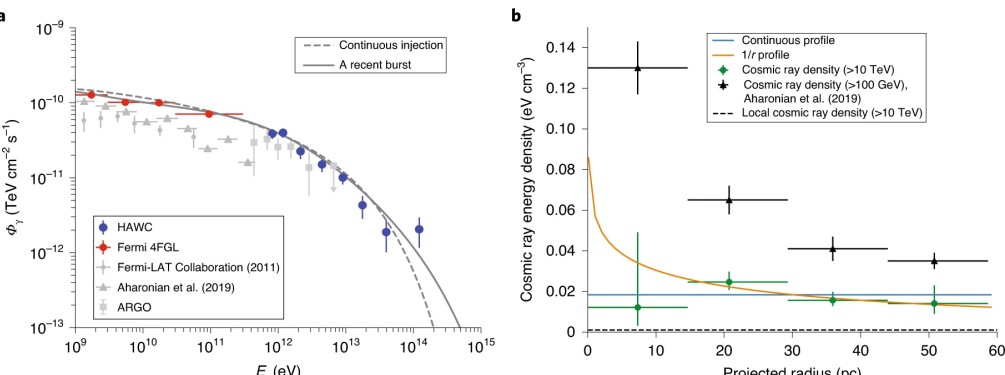

**Figure 10.** (**a**) Figures from [64]. Spectral energy distribution of the gamma-ray emission at the Cocoon region. HAWC errors on the flux points are the 1 sigma statistical errors. At low TeV energy, HAWC data agree with the measurements by the ARGO observatory [88]. The red and grey circles are the Fermi-LAT flux points published in [87,89], respectively. The grey triangles are from the Fermi-LAT analysis in [44]. The grey solid and dashed lines are the spectra derived from the hadronic modeling of the region. (**b**) Cosmic ray energy density profile calculated for four rings (0–15 pc, 15–29 pc, 29–44 pc, and 44–55 pc) centered at the OB2 association. The green circles are the cosmic ray densities derived above 10 TeV using HAWC gamma-ray data. The y errors are the statistical errors and the x error bars are the width of the x bins. The orange and blue lines are the 1/r profile for the case where the particles are continuously injected) and constant profile (signature of the burst injection), respectively. The black dashed line is the local cosmic ray density above 10 TeV, based on the Alpha Magnetic Spectrometer measurements. The black triangles are the cosmic ray densities above 100 GeV from [44].

## 4. Conclusions

Two decades ago, only a few sources of very high energy gamma rays had been detected, and few astrophysical sources were thought to be able to accelerate particles to very high energies within the Galaxy. Before the advent of Fermi-LAT in 2008, the EGRET source catalogue counted 288 sources [90]. An order of magnitude difference in sensitivity between the previous and current generation of instruments has revolutionized gamma-ray astronomy. From about 10 instances of TeV sources in 2002 before the advent of the H.E.S.S., VERITAS, and MAGIC instruments (http://tevcat.uchicago.edu/, accessed on 14 September 2022), the source catalogues nowadays contain thousands of astrophysical objects emitting at GeV energies, and hundreds of objects emitting up to TeV energies, some of them up to hundreds of TeV and even PeV gamma rays (https://heasarc.gsfc.nasa.gov/W3Browse/fermi/fermilpsc.html and http://tevcat.uchicago.edu, accessed on 14 September 2022). Not only has the number of detected sources dramatically increased, but also the variety of astrophysical phenomena that emit high energy gamma rays is

amazing. Observational evidence shows that the efficient acceleration of particles occurs very often in Galactic astrophysical sources. SNRs accelerate particles up to 100 TeV. However, the total power of the supernova explosions injected into the CRs is unknown, and we do not know whether the bulk of the Galactic CRs is accelerated in SNRs. This is due partly to the degeneracy of the hadronic and leptonic emission mechanisms, which could be solved if we knew the gas, magnetic, and radiation densities in SNR shells, partly to particles escaping the remnants. Nor do we have observational evidence that SNRs are the Galactic factories of PeV CRs, as the highest energy CRs run away so early that it is extremely challenging to detect them within the SNR shells.

In the last few years, the hunt for the most extreme CR factories in the Galaxy, the PeVatrons, has become a major focus in high energy astrophysics. The solution to the question on the maximum acceleration energy achieved in the Galaxy and on the major contributors to Galactic CR flux has to come necessarily from the high-significance spectra of multi-TeV to PeV photons produced by CRs close to the knee energy colliding with the ambient gas. The spectrum of the highest energy gamma-ray radiation from different PeVatron candidates contains crucial information on the source acceleration mechanisms, and on the contribution of the different sources to the formation of the knee in the local CR spectrum.

Unbiased surveys of the Galactic Plane in the crucial multi-TeV up to PeV energy range are currently being undertaken with the high altitude water Cherenkov detector, HAWC [34,63,64], and the Large High Altitude Air Shower Observatory, LHAASO [65], and the Tibet Array [69,91]. In the future, similar searches will be undertaken by the Southern Wide-field Gamma-ray Observatory (SWGO) [92], and by the upcoming Cherenkov Telescope Array [93], which will benefit from high angular resolution, which is helpful for comparing the gamma rays and gas maps of the Galaxy.

Hadronic collision of protons off the ambient gas also produce X-rays, the result of the synchrotron radiation of secondary electrons, the products of $\pi^{\pm}$-decay. The X- and gamma-ray fluxes depend on the present content of the total energy of accelerated protons accumulated in the CR source, and on the density of the ambient matter. Approximately the same fraction of energy of the parent protons is transferred to secondary electrons and gamma rays. The lifetime of these secondary electrons, $t_{synch}1.5(B/mG)^{3/2}(E_X/1keV)^{-1/2}$ year, is very short ($\leq 50$ years) [94]. In the search for the proton PeVatrons, these X-rays from secondary electrons could be treated as a *prompt* radiation emitted simultaneously with gamma rays, a valuable source of complementary information to constrain the acceleration efficiency of the PeVatron candidates. The $L_X/L_\gamma$ ratio depends on the proton spectrum, as well as on the particle injection history; for a $-2$ proton spectrum, the ratio does not exceed 0.2–0.3. [94]. Thanks to its field of view, (roughly 1 degree), which is wide if compared to other X-ray instruments, eRosita (https://www.mpe.mpg.de/eROSITA, accessed on 14 September 2022) has conducted a survey of the X-ray sky. This survey is a very valuable set of data for searching for secondary X-ray emissions from GeV–TeV sources, which are usually extended. More generally, this uniquely wide field of view X-ray instrument can be used to constrain the electron population and magnetic fields in the regions of extended GeV–TeV sources.

**Funding:** This research was funded by the Polish Science Centre, grant number DEC-2017/27/B/ST9/02272.

**Conflicts of Interest:** The author declares no conflict of interest.

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
