# Peer review of "On the Search for the Galactic PeVatrons by Means of Gamma-Ray Astronomy"

_universe, doi:10.3390/universe8100505_

Round 1

Reviewer 1 Report

The paper presents a good review of the possible acceleration sites of cosmic rays using data on the gamma and X-ray emissions from different detectors like HESS, FERMI, LHASO, and HAWC. The paper is well-written and understandable. Hereafter are some points that have to be addressed, mainly about the resolution of the figures and how they are presented.

1) Page 1 line 1: "Cosmic rays are ultra-relativistic electrons and protons" -> Cosmic rays are ultra-relativistic particles...

2) Page 2 line 89: it is not clear to me from figure 1 (radial profile) and its description, how to identify the data for the three different energy (<1 TeV 1<TeV<3 >3TeV).

3) Figure 1: It seems there is no description of what the fitting curve represents.

4) page 3 line 95: typo "Tt is not..."

5) Page 3 line 98-100: repetition in the same sentence "Fig 2 Fig 2"

6) Figure 2: on this picture as in other pictures is not present the reference to the paper from where the pictures have been taken. I would suggest also adding the name of the SNR to the caption of the figure.

7) Page 5 lines 173 -175: it seems the quantity We Wp n wMBR and WB has not been defined.

8) page 7 line 1986-197: I think it would be better to define IACTs. "(see 3)" -> (see Fig. 3)

9) Figure 6: in the caption, the supernova is called SNR G338.3-0.6 but in the picture, it seems it is SNR G338.3-0.0.

10)Figure 7 and Figure 8: the resolution of these two pictures is extremely bad, I think a better resolution is needed. Moreover, it seems they are not cited in the text.

11) Figure 9: also this picture needs a better resolution.

Reviewer 2 Report

In the refereed manuscript the Author reviews gamma-ray observations of Galactic cosmic ray sources and related theoretical interpretations concerning the nature of these sources (for instance, hadronic vs. leptonic sources). I have a number of comments, suggestions, and questions, which I list below. I am sorry for the delay with this report --- the text of the report is longer than I could initially conceive; this (in part) caused the delay.

1) The Author aims to “discuss how gamma-ray observations of different astrophysical sources can constrain the sites and acceleration mechanisms producing the hadronic part of the cosmic ray flux we measure at Earth” (lines 54-56). The main interest of the Author is “understanding the acceleration mechanisms of the fast perfect accelerators capable of injecting particles up to PeV energies” (lines 58-59). Furthermore, in the abstract the Author argues that “Gamma-rays emitted when cosmic rays interact with the Galactic gas and radiation fields are a powerful tool to investigate their origin”.

As the paper stands, it appears that, at present, this great promise of gamma-ray astronomy does not always live up to such great expectations. Namely, a robust observation of a sizable population of hadronic PeVatrons that actually could produce the knee observed at Earth seems to be beyond what is currently possible (please find some details below, point 4). If the Author agrees with this last statement, probably it is worth including this or similar statement to the text; otherwise, the enthusiastic reader could be somewhat disheartened towards the end of the reading.

In addition, it is perhaps justifiable to change the rationale of the paper from reviewing results that have supposedly identified the hadronic PeVatrons --- knee makers” to a more reserved discussion of both hadronic and leptonic mechanisms and identifying the signatures of the hadronic PeVatrons/TeVatrons that could be hopefully more robustly found in future observations. Per se, it would not change the text of the manuscript significantly since both hadronic and leptonic mechanisms are already discussed in the manuscript.

2) lines 38-43 “For CRs from GeV to PeV energies the dominant emission processes are the decay of neutral pions, \pi0 , produced when CR hadrons collide with ambient gas in the ISM, which is commonly called the hadronic production mechanism, and inverse Compton (IC) scattering of CR electrons off radiation fields, which is called the leptonic production mechanism. Either process will produce gamma rays of around 10% of the energy of the parent cosmic ray”

To my understanding, the average energy of the secondary gamma-ray resulting from inverse Compton scattering is rarely strictly proportional to the energy of the primary electron. Rather, in the Thomson regime E_{gamma-sec} \propto E_{e-prim}^{2}; in the extreme Klein-Nishina regime E_{gamma-sec} \approx E_{e-prim}. I would appreciate it very much if the Author could give a more physically motivated description of the inverse Compton process with requisite references and, if necessary, could identify specific cases when, indeed, E_{gamma-sec} could be approximately proportional to E_{e-prim} (please see also lines 169-170 and other possible places in the text where this could be important).

3) Likewise, the manuscript contains a number of statements that are sometimes correct given a specific astrophysical system / theoretical model, but do not always prove correct. Sometimes these statements go in not-so distant, but apparently somewhat contradictory pairs:

lines 21-22: “The knee is thought to mark the change from Galactic into extra-Galactic CRs” and lines 27-28: “Up to knee energies and maybe as high as 10 17 − 18 eV in energy CRs are believed to have a Galactic origin”

Comment: if cosmic rays at 10^{18} eV are still Galactic, the knee could hardly mark the change from Galactic into extra-Galactic CRs even if these CRs are exclusively composed of heavy (say, Iron) nuclei. The apparent confusion could be probably somewhat allayed if the Author could introduce at least some very basic concepts of CR physics, such as the rigidity-dependent cutoff (e.g. Hoerandel 2003; Antoni et al. (KASCADE) 2005; works of KASCADE-Grande) and the possibility of the “second” Galactic component beyond the knee. Or, alternatively, it is possible to simply write that CR protons up to at least several PeV could be of a Galactic origin.

lines 42-43: “Either process will produce gamma rays of around 10% of the energy of the parent cosmic ray” and lines 43-46: “Typically, the spectrum of the hadronic radiation at TeV mimics the parental CR spectrum where the proton energy is shifted to lower energy by a factor 20-30, so that PeV CRs emit γ-rays at hundreds TeV.”

Comment: I believe that for the hadronuclear (mainly pp) process it is probably sufficient to introduce a fixed factor between the energy of the primary proton and the average (or, alternatively, most probable, or median) energy of the secondary gamma-ray. If you wish to consider the photohadronic (mainly p gamma) process,  could you please state that explicitly and introduce the separate factor between the energies of the proton and the gamma-ray.

lines 123-125: “SNRs are believed to accelerate the highest energy CRs (up to at least a few PeVs) at the transition between the free expansion and the Sedov phase, within few hundred years after the supernova explosion”

and lines 134-137: “Very young SNRs are clearly primary targets to test the SNR paradigm. The youngest known SNR in the Galaxy is G1.9+0.3, which is only 110 yr old. While its shock speed of 10000 km/s, estimated through X-ray measurements, is ideal to produce efficient diffusive acceleration”

Comment: apparently these is some confusion as to whether the best target for observation is 1) the youngest SNR or 2) a few-hundred-year-old SNR. Could you please elaborate on that. If this is not clear could you please give a (more or less complete) list of factors in favor of 1) and 2) (please see also lines 332-333 --- there the argument seems to be in favor of the youngest SNR)

lines 354-358:“However, since the energy of sub-TeV electrons is not radiated away effectively, the direct (π 0 -decay) gamma-ray luminosity exceeds the synchrotron luminosity. The L X /L γ ratio depends on the proton spectrum as well as on the particle injection history; typically it does not exceed 0.2-0.3. In the search for the proton PeVatrons these X-rays from secondary electrons could be treated as a prompt radiation (...)”

Comment: it is hardly possible that “the energy of sub-TeV electrons is not radiated away effectively” and “secondary electrons could be treated as a prompt radiation”. If the radiation is prompt, the energy is usually radiated rather effectively. Probably there is some transition between electron energies/regimes here, but this transition is somewhat hidden. Could you please elaborate on that.

4) Concerning the search for hardonic PeVatrons accelerating protons up to the knee (without cutoff/break up to that energy):

a) In Section 2 it is shown that known SNRs do not solve the problem

b) The Galactic Centre (Subsection 3.1) is, indeed, a promising PeVatron candidate; however, it is not clear whether a sufficient population of multi-PeV protons accelerated there could be “transferred” near the Earth to create the local CR “fog”. This is dependent on the structure of magnetic field in the Galaxy and, in particular, near the Galactic Centre. One could naively expect that most multi-PeV protons would leave that Galactic disk and would not return inside it.

c) Many LHAASO sources reveal a break/cutoff as shown in Fig. 8.

d) Even if we see “clumps” of gamma-radiation, this could be due to filaments accelerating electrons. Meanwhile, I am in doubt that the limited angular resolution could allow to link these clumps (if indeed produced by protons/nuclei) with concentrations of gas. Please also consider a sizable systematic pointing uncertainty.

e) Could diffuse gamma-rays (Tibet-AS\gamma, and now also LHAASO) be due to leptonic sources? In principle yes, if there is a large population of very faint leptonic PeVatrons in the Galaxy.

I would appreciate it very much if the Author could state explicitly for which sources we could be reasonably well sure that these are hadronic PeVatrons accelerating protons to the knee and creating the local CR fog (i.e. the leptonic hypothesis is rejected at the statistical significance at least 5 \sigma). If there is no such source(s), it would be helpful to state this also.

5) lines 121-122: “The short-time variability likely tells us that hadrons together with leptons are accelerated in the shocks.”

Could you please elaborate on that; unfortunately, I could not understand the precise mechanism of how the fast variability leads to the efficient acceleration of protons or nuclei.

6) lines 211-213: “The gamma-ray distribution follows closely the molecular gas distribution and can be interpreted as produced by hadrons.”

did you implicitly assume here that the diffusion coefficient is independent of the radius: D(R)= const? If so, could you please provide a justification for this assumption.

7) lines 223-225: “(…) in this case the parental proton spectrum should extend to energies close to the knee. In Fig 6 we show how the parental particle population can be deduced from the TeV emission.”

Probably one could agree with this statement if we assume the primary spectrum as a “blunt” (i.e. not sharp) knee or a power-law spectrum with an exponential cutoff (I apologize, but I could not elucidate the precise shape of the primary spectrum; could you please add this to the text or indicate where it is given). But what if we have the primary proton spectrum as a sharp knee? I would naively expect that it could be hard to set such stringent lower limits on the proton cutoff energy given that the maximal gamma-ray energy in Fig. 6 is only about 20-30 TeV in both cases. Could you please elaborate on that.

8) lines 291-293: “The 1/r decrement of the CR density with the distance from the star cluster is, in fact, a distinct signature of continuous injection of CRs over the cluster lifetime and their diffusion through ISM.”

again, does not the gamma-ray profile depend on D(R) (please see point 6) above)? If we don’t know D(R), how could we derive the profile of the cosmic ray number density vs. the radius R?

Minor comments:

1) The title of the review “On the sources of the highest energy particles in the Galaxy” could make one think that this is a review on the Galactic --- extra-Galactic cosmic ray transition (that probably takes place between 10^{17} eV and 10^{19} eV). Please consider changing the title to “On PeVatrons in the Galaxy observed by means of gamma-ray astronomy” or something similar.

2) Abstract: “Cosmic rays are ultra-relativistic electrons and protons”. Please consider changing to “Cosmic rays are ultra-relativistic electrons, protons, and nuclei”. Heavy nuclei could be important between the energy of 10 PeV and 500 PeV in cosmic ray physics, even if they are much less important in gamma-ray astronomy.

3) Abstract: “We give here a state of the art report on the search for the sources of Galactic cosmic rays”. Please consider changing to “We give here a state of the art report on the search for the sources of Galactic cosmic rays by means of gamma-ray astronomical methods” or something similar.

4) lines 31-33. “Since CRs are charged particles, they are deflected in the Galactic magnetic fields and isotropised, so that it is impossible to directly observe them close to their suspected acceleration sites and thus pin down their sources.”

Did you mean something like “(…) close to the direction from their suspected acceleration sites”, i.e. that CRs do not point to the sources because they deflect in magnetic fields? If so, maybe it is better to state this explicitly.

5) lines 101-105: The full-remnant spectral energy distribution at gamma-ray energies shown exhibits a hard spectrum in the GeV regime, a flattening between 100 GeV and a few TeV, and an exponential cutoff above 10 TeV. In a hadronic scenario, this spectrum can be fitted with a broken power law proton spectrum with a break at 1.4 ± 0.5 TeV and an exponential cutoff at few TeV.”

Probably I have misunderstood something, but if the gamma-ray cutoff is at ~10 TeV, then I would expect the primary proton cutoff at ~100 TeV, not at “few TeV” (a few TeV?). Please confirm.

6) lines 235-239: “The photon energies reported by the LHAASO Collaboration are so high that the attenuation of these high-energy gamma-rays, due to pair production of gamma-rays interacting with background photons from both the cosmic microwave background (CMB) and interstellar radiation fields (ISRF), has been taken into account for the first time for radiation sources located within the Galaxy.”

If I remember it correctly, the effect itself is known at least from 2006 (or even 2004-2005) (could you please add these early references if you consider it possible/advisable). Probably it is more precise to say that this effect revealed itself only in LHAASO data due to the capability of LHAASO to observe at the highest energies, detect relatively distant sources, and measure the gamma-ray spectrum with relatively small statistical uncertainties.

7) lines 314-324:“A decade ago only few sources of very high energy gamma-rays had been detected and few astrophysical sources (...) Not only has the number of detected sources overtaken the most optimistic predictions before the advent of the current generation of gamma-ray detectors”

Could you please give here more specific numbers. For space gamma-ray telescopes probably it is better to count since the Fermi-LAT launch (2008) rather than from 2012. The same for ground-based instruments: it would be great to find a specific suitable “delimiting” year to demonstrate the fast progress here. Could you please be more specific about how “the number of detected sources overtaken the most optimistic predictions” (which predictions these actually were).

8) lines 345-347: it would be helpful if the Author could include some references describing the performance of Tibet-AS\gamma, and the expected performance of SWGO and the Cherenkov Telescope Array.

9) lines 360-364: "Thanks to its wide field of view the X-ray instrument, eRosita (https://www.mpe.mpg.de/eROSITA), is a very valuable instrument for searching for secondary X-ray emission from GeV-TeV sources, which are usually extended. More generally, this unique wide field of view X-ray instrument can be used to constrain the electron population and magnetic fields in the regions of extended GeV-TeV sources."

Due to limited time, I failed to find a specific value (sorry for that!), but from a presentation by Prof. Sunyaev I have conceived that the field of view of eRosita is actually not so wide (probably ~1 degree) if compared to e.g. Fermi-LAT. Could you please check that and if it is so to be more specific about the numbers. Did you mean that the field of view of eRosita is wider than for other X-ray telescopes (not Fermi-LAT)? Than could you please state that explicitly (with requisite numbers).

Very minor points:

line 52: “These X-rays from secondary electrons can treated as a prompt radiation” did you mean “can be treated”?

line 60: “In this contest gamma-ray observations are the best tool” did you mean “In this context”?

line 95: “Tt is not clear” did you mean “It is not clear”?

lines 98-100: “As evident in Fig.2 both hadronic and leptonic emission model can account for the broadband radiation from the shell of RX J1713.7-3946 as shown in Fig.2 (leptonic-hadronic degeneracy).”

As evident in Fig. 2 … as shown in Fig. 2” probably this repetition is not strictly necessary

lines 179, 180, 181: “10 Ì„G” --- this is probably a misprint; please correct.

To conclude, I would like to encourage the Author to undertake a major revision of the manuscript.

Round 2

Reviewer 2 Report

I am grateful to the Author of the manuscript for the work made. I believe that the manuscript could be accepted for publication in Universe.